# Probiotic Properties of Lactic Acid Bacteria with High Conjugated Linoleic Acid Converting Activity Isolated from *Jeot-Gal*, High-Salt Fermented Seafood

**DOI:** 10.3390/microorganisms9112247

**Published:** 2021-10-28

**Authors:** Nho-Eul Song, Na-Jeong Kim, Young-Hun Kim, Sang-Ho Baik

**Affiliations:** 1Department of Food Science and Human Nutrition, Jeonbuk National University, Jeonju 54896, Korea; nesong@kfri.re.kr (N.-E.S.); najung1230@nate.com (N.-J.K.); 2Department of Animal Science and Institute of Agricultural Science & Technology, Jeonbuk National University, Jeonju 54896, Korea; ykeys2584@snu.ac.kr

**Keywords:** conjugated linoleic acid, lactic acid bacteria, *Jeot-gal*s, probiotics, linoleic acid isomerase

## Abstract

Conjugated linoleic acid (CLA) isomers are potent health-promoting fatty acids. This study evaluated the probiotic properties of 10 strains of high CLA-producing lactic acid bacteria (LAB) isolated from *Jeot-gal*, a high-salt, fermented seafood. Two isolates, *Lactiplantibacillus plantarum* JBCC105683 and *Lactiplantibacillus pentosus* JBCC105676, produced the largest amounts of CLA (748.8 and 726.9 μg/mL, respectively). Five isolates, *L. plantarum* JBCC105675, *L. pentosus* JBCC105676, *L. pentosus* JBCC105674, *L. plantarum* JBCC105683, and *Lactiplantibacillus paraplantarum* JBCC105655 synthesized more *cis*-9, *trans*-11-CLA than *trans*-10, *cis*-12-CLA (approximately 80:20 ratio). All the strains survived severe artificial acidic environments and showed antimicrobial activity and strong adhesion capability to Caco-2 cells as compared to the commercial strain *Lactocaseibacillus rhamnosus* GG. Among them, *Pediococcus acidilactici* JBCC105117, *L. paraplantarum* JBCC105655, and *L. plantarum* JBCC105683 strongly stimulated the immunological regulatory gene PMK-1 and the host defense antimicrobial peptide gene *clec-60* in *Caenorhabditis elegans.* Moreover, three strains showed a significant induction of tumor necrosis factor-α, interleukin (IL)-1β, IL-6, IL-12, and IL-10 production in RAW 264.7 macrophages, indicating that they were promising candidates for probiotics with high CLA-converting activity. Our results indicate that the newly isolated CLA-producing LAB might be useful as a functional probiotic with beneficial health effects that modulate the immune system.

## 1. Introduction

*Jeot-gal*s are traditional Korean fermented, high-salt seafoods prepared from various fishes, shrimps, oysters, fish eggs, and fish intestines with final salt concentrations of up to 30% (*w*/*w*) [1,2]. During fermentation for several months, or several years for some *Jeot-gal*s, the unique microbial consortium results in lots of aerobic and anaerobic microorganism developments, including lactic acid bacteria (LAB) during the maturing period [3]. LAB account for 70~90% and consist mainly of the genus *Lactobacillus* and the genus *Weissella* [4]. Many studies have shown that functional strains, such as halophillic and halotolerant bacteria [5,6,7,8] and γ-aminobutyric acid or bacteriocin isolated from *Jeot-gal*s, are indications of useful microbial sources of LAB [9,10]. Hence, the LAB from *Jeot-gal*s were expected to possess some novel and useful properties as marine environments are different from terrestrial environments.

CLA is a mixture of the positional and geometric isomers of linoleic acid (LA, *cis*-9, *cis*-12-octadecadienoic acid) synthesized naturally in the human colon. It usually exists in two biologically active isomers, *cis*-9, *trans*-11-CLA and *trans*-10, *cis*-12-CLA, which have different potential health benefits [11,12]. The isomer *cis*-9, *trans*-11-CLA primarily exhibits enhanced immunity [13] and anticarcinogenic [14], antiatherogenic [15], antidiabetic [16], and anti-obesity activity [17,18]. The isomer *trans*-10, *cis*-12-CLA exhibits activities that enhance energy metabolism, reduce lipid content [19,20,21], promote bone health [22], and improve growth performance [23] in animal models and in various diseased human cell lines. The conversion of LA to CLA occurs in the rumen during biohydrogenation through several beneficial intestinal bacteria. Although CLA isomers naturally exist in several food products derived from ruminants, the amounts of CLA are relatively small in these foods, which require large consumption for beneficial health effects [24]. Various lactic acid bacteria (LAB), particularly the *Lactobacillus* species, have been used to produce high-purity *cis*-9, *trans*-11-CLA and *trans*-10, *cis*-12-CLA through enzymatic isomerization [21]. Nevertheless, there is no information about LAB having the ability to produce conjugated linoleic acid (CLA) from *Jeot-gal*s or its probiotic properties.

Food-grade CLA-producing bacterial strains added during manufacturing may increase the content of this beneficial compound in fermented products [25]. Hence, a high conversion activity of LA to CLA is required to increase the supply of these fatty acids in the human colon. Additionally, the production of these bioactive fatty acid metabolites may be considered a probiotic trait. Thus, the strains that produce bioactive CLA and show the characteristics of probiotics may have beneficial effects on health. Recently, in the food industry, consumer demand has increased for symbiotics that improve well-being through nutraceutical functions in addition to probiotic properties. These so-called symbiotics contain biologically active compounds that exert beneficial health effects and decrease the risk of certain diseases with probiotic properties [26]. To be considered a probiotic, a bacterial strain should survive under the extreme conditions of the gastrointestinal (GI) tract (low pH and presence of bile salts), adhere to the intestinal epithelial cells, and exert beneficial effects on the host, including antimicrobial and immunomodulatory properties [27].

A recent study reported that strain-specific, immune-regulatory effects on the host are important because microbiota imbalance leads to inflammatory bowel diseases through the overexpression of pro-inflammatory cytokines [28]. Additionally, in vivo, LAB may contribute to host defense and prolong the lifespan of *Caenorhabditis elegans*, which is an important experimental animal in innate immunity research due to its suitability for genetic analysis and reproducible life span [29]. However, the relationship between CLA-producing probiotic bacteria and the immune regulation in *C. elegans* has not been reported. The immune system of the intestinal barrier consists of a combination of mucus, intestinal epithelial cells, immunoglobulin A (IgA), antimicrobial peptides, and immune cells. This barrier protects the gut microbiota by maintaining it in the intestines through communication between the immune system and the microbiota [30]. Recently, a *C. elegans* surrogate in vivo model, as an alternative to in vitro models, such as those propagated in human intestinal cell lines, has been successfully used as a simple, rapid, and economic model system to study bacteria–host interactions in the gut because the intestinal cells of *C. elegans* are similar in structure to those in humans [31]. The nematode innate immune system is highly conserved and can be quickly activated through several immune-regulatory pathways to protect the host [17]. Among these, the p38 MAPK and DAF/IGF signaling pathways play a vital role in combating bacterial infection in the intestine. In particular, the NSy-1-SEK-1-PMK-1 cassette of p38 MAPK is a central regulator of host defense in *C. elegans* and depends on the upstream scaffold protein TIR-1 for its activity. The exposure of *pmk-1*::*GFP* to selected LAB strains indicates LAB strains that might positively correlate with the host defense system [32,33]. *clec-60* is a gene encoding antimicrobial peptide that is regulated by PMK-1 signaling and can be used to evaluate the impact of conditioning with LAB strains on *clec-60* associated with nematode immune responses through qRT-PCR analysis.

The aim of this study was to isolate novel CLA-producing LAB from *Jeot-gal*s, which are a useful source for isolating probiotic strains with high cell viability under low pH conditions, and to assess their properties to determine their potential as probiotics. Thus, we evaluated the probiotic activities of these strains, including their ability to survive under extreme conditions, their antimicrobial activity, their adhesion to human intestinal epithelial cells, and their immune modulation in vivo by *C. elegans* as well as in vitro by inflammatory cytokine assay.

## 2. Materials and Methods

### 2.1. Chemicals and Media

All the standard samples of LA, CLA (*cis*-9, *trans*-11-CLA and *trans*-10, *cis*-12-CLA), heptadecanoic acid as an internal standard, and the antibiotics were purchased from Sigma-Aldrich (St. Louis, MO, USA). The de Man, Rogosa, and Sharpe (MRS) medium for LAB cultivation and nutrient broth, brain heart infusion broth, and the PP medium for pathogenic microorganisms were obtained from BD Difco Laboratories (Detroit, MI, USA). All the other chemicals and reagents used were of analytical grade. *Lactocaseibacillus rhamnosus* GG (ATCC53103) was used as a control for the probiotic characterization.

### 2.2. Screening and Identification of LAB with LA to CLA Conversion Activity

Rapid bacterial screening for the LAB converting of LA to CLA was conducted in two steps: (1) LA resistance [34], with some modifications, and (2) UV spectral scan analysis. First, approximately 10 g of *Jeot-gal* samples were homogenized at 500 rpm in a rotor-stator homogenizer (U-TRON LAB/P, ESYNDMT, Sasang-gu, Busan Korea) and serially diluted 10-fold with saline (0.85%). The diluted homogenized samples were plated on MRS agar plates containing 0.2% (*v*/*v*) LA with 1% (*v*/*v*) Tween 80. After the inoculated plates were incubated at 30 °C for 48 h, only strains showing large and well-formed colonies were selected for the next steps. The selected strains were cultured in 5 mL MRS broth containing LA (5 mg/mL) with 1% (*v*/*v*) Tween 80 and scanned using a UV/Visible spectrophotometer (Beckman Coulter, Brea, CA, USA). Second, fatty acids in the culture supernatant were extracted using isopropanol (99.9%)/hexane extraction to confirm the conversion of CLA from LA by UV spectral scan analysis, as described by Liu et al. [11]. A 1 mL sample and 2 mL of isopropanol were mixed vigorously for 1 min. Next, 1.5 mL of hexane was added and vortexed for 2 min at room temperature. The upper layer obtained after centrifugation (4000× *g* for 5 min) was used for downstream analysis. The fatty acid extraction samples obtained above were scanned from 200 to 300 nm using a Beckman DU 800 UV/Visible spectrophotometer (Beckman, Fullerton, CA, USA) according to Liu et al. [11]. A characteristic absorption peak at 228–235 nm was expected, indicating the possible presence of CLA with conjugated double bonds in the extraction samples. Absorbance was measured in 1 cm quartz cuvettes at room temperature, and spectral graph data and absorbance values of the samples were obtained. For identification of the obtained strains, 16s rRNA sequencing was used as described by Kim and Baik [35].

The CLA isomers were methylated for GC analysis and quantified using methyl esters prepared by acid-catalyzed methylation methods by gas chromatographic analysis. The hexane solvent in the extracted samples was dried with nitrogen gas, and then the residue was added to 1 mL of the internal standard solution prepared by dissolving heptadecanoic acid in isooctane (2,2,4-trimethylpentane) to produce a final concentration of 1 mg/mL. After 4 mL of 0.5 N sodium methoxide (CH_3_ONa, 2 g NaOH in 100 mL methanol, *w*/*v*) was added, the sample was heated at 80 °C for 5 min and cooled to room temperature. Next, 5 mL of 14% boron trifluoride-methanol (BF_3_/CH_3_OH; *w*/*v*) was added, and the sample was heated at 80 °C for 10 min. After cooling to room temperature, the methylated sample was extracted with 3 mL of isooctane by vortexing for 30 s. Subsequently, 5 mL of saturated sodium chloride was added and vortexed for 30 s. The upper layer was dried over anhydrous sodium sulfate and analyzed by GC (Hewlett-Packard 5890, Agilent Technologies Inc., Santa Clara, CA, USA) equipped with a flame ionization detector and a Supelco SP-2560 fused silica capillary column (100 m × 0.25 mm, i.d., 0.2 μm film thickness; Supelco Inc., Bellefonte, PA, USA). Fatty acid methyl esters were separated as described by Beppu et al. [36]. Heptadecanoic acid was used as an internal standard, and the CLA peak was identified by comparing the retention time to the standards.

### 2.3. Acid and Bile Tolerance of Isolated Strains

Each LAB (8–9 log colony-forming units [CFU]/mL) was harvested by centrifugation (4000× *g* at 4 °C for 10 min) and suspended in an equal volume of the MRS broth, which was adjusted to pH 2.0 with 5.0 M HCl to investigate acid tolerance. After incubation at 37 °C for up to 2 h, the survival rate was evaluated by determining the viable cell counts of the samples after serial dilution in sterilized water and incubation at 37 °C for 48 h. For bile tolerance, 1% of the cell suspension was inoculated onto plates supplemented with 0, 0.3%, 1%, 3%, and 5% (*w*/*v*) oxgall (Sigma-Aldrich, St. Louis, MO, USA), and the survival rate was evaluated by determining the viable cell counts as described above.

### 2.4. Antimicrobial Activity

The antimicrobial activities against various bacterial pathogens were evaluated against pathogenic microorganisms, such as *Staphylococcus aureus* KCTC 1916, *S. epidermidis* KCTC 1917, *S. xylosus* KACC 13239, *Pseudomonas aeruginosa* KACC 10186, *P. putida* KACC 10266, *Bacillus cereus* KACC 10097, *B. subtilis subsp. spizizenii* KACC 14741, *B. vallismortis* KACC 12149, *Escherichia coli* KACC 13821, and *Propionibacterium acnes* KCTC 3314, in order to determine their inhibitory capacity against intestinal opportunistic pathogens, as described by Kim and Baik [35]. 

### 2.5. Cell Adherence Assay 

The adherence assay of LAB to Caco-2 cells in vitro was carried out as previously described [35], with minor modifications. The Caco-2 intestinal epithelial cells (5 × 10^4^ cells/well) were cultured in MEM (Lonza, Basel, Switzerland) containing 10% heat-inactivated fetal bovine serum (Gibco, Grand Island, NY, USA) in a 5% CO_2_ humidified incubator (NU-5800; NuAire, Caerphilly, UK). The Caco-2 cell monolayers were incubated with LAB (8–9 log CFU/mL) for 2 h at 37 °C and washed with PBS to remove unadhered bacteria and treated with 0.1 mL of 0.05% (*v*/*v*) Triton X-100 to extract the adhered bacteria. The cells were then serially diluted, plated on MRS agar, and incubated for 48–72 h at 37 °C for bacterial adhesion calculation based on the number of viable bacteria in the original suspension and the cell lysates and expressed as log CFU/mL. 

### 2.6. In Vivo Caenorhabditis Elegans Assay for Immune Response

For the in vivo assay of the LAB, the *C. elegans* CF512 *fer-15(b26)II;fem-1(hc17)IV* (*fer-15;fem-1* worms) strain was routinely maintained on nematode growth medium (NGM) plates seeded with *Escherichia coli* OP50. The LAB were sub-cultured (9 log CFU/mL) three times before use and exposed to the *C. elegans* on NGM plates containing nystatin for 5 days. Ten worms were randomly picked, washed twice with M9 buffer, and placed on brain heart infusion plates containing kanamycin and streptomycin. After exposure to gentamycin (5 μL of a 25 μg/mL solution) for 5 min, the worms were washed three times with M9 buffer and then pulverized using a pestle (Kontes Glass Inc., Vineland, NJ, USA) in a 1.5 mL Eppendorf tube containing M9 buffer supplemented with 1% Triton X-100. After serial dilution in the M9 buffer, the worm lysate was plated on the MRS agar (pH 5.0) plates, incubated for 48 h at 37 °C, and then counted for live bacterial cells. The results were compared with those obtained using *L. rhamnosus* GG as a positive control. To elucidate the transcriptional host responses [*pmk-1*::*GFP* in wild-type N2], the nematodes were induced by exposure to selected LAB strains for 24 h. The animals were mounted on glass slides with 2% agarose pads, anesthetized with 10 mM NaN_3_, and quickly visualized using an AxioImager Z1 fluorescence microscope (Zeiss, Oberkochen, Germany). Quantitative reverse transcription polymerase chain reaction (qRT-PCR) analysis was performed to evaluate the impact of conditioning with the LAB strains on *clec*-60, a gene associated with nematode immune responses. Transcript levels were measured in young adult *C. elegans* N2 Bristol wild-type, conditioned with the LAB strains for 24 h. The total RNA from the worms was quickly isolated following the TRIzol (Invitrogen, Waltham, MA, USA) method and purified using the RNeasy Mini Kit (QIAGEN, Germantown, MD, USA), including an on-column DNase digestion with RNase-free DNase (QIAGEN, Germantown, MD, USA). After RNA isolation, the total RNA (50 ng) was used for quantitative real-time PCR (qRT-PCR) using the SuperScript III Platinum SYBR green one-step qRT-PCR kit (Invitrogen, Waltham, MA, USA). The qRT-PCR was performed using the StepOne™ Real-Time PCR System (Applied Biosystems, Foster City, CA, USA). Primers were designed using Primer3Input software (v0.4.0) and are listed as follows: *clec-60* (5′-ACGGGCAAGTTATTGGAGAG-3′ and 5′-ACACGGTATTGAATCCACGA-3′) and *snb-1* (5-CCGGATAAGACCATCTTGACG-3′ and 5-GACGACTTCATCAACCTGAGC-3. The control gene *snb-1* was used to normalize the gene-expression data.

### 2.7. qRT-PCR for Inflammatory Cytokine Assay 

RAW 264.7 cells (5 × 10^4^ cells/well) were seeded into 96-well plates and heat-killed LAB (100 °C, 30 min) suspended in PBS (pH 7.4) were added to the wells at a final concentration of 1 × 10^8^ cells/mL. The effects of the strains on immune cell proliferation were determined using the MTT assay. RAW 264.7 cells were stimulated with 1 mg/mL lipopolysaccharide (LPS) and the LAB. Total cellular RNA was extracted from RAW 264.7 cells using a NucleoSpin RNA kit (Macherey-Nagel, Düren, Germany). The total RNA (500 ng) was reverse-transcribed to cDNA using a Quick reverse transcription system (ReverTraAce qPCR RT Master Mix; TOYOBO, Osaka, Japan). The cDNAs were amplified by PCR using SYBR Green Mastermix (TOYOBO, Osaka, Japan) and specific primers (Appendix A). The qRT-PCR was performed as follows: initial denaturation at 95 °C for 2 min, 39 cycles at 95 °C for 15 s, 57 °C for 20 s, and 72 °C for 30 s (CFX96 Touch^TM^ Real-Time PCR Detection System, Bio-Rad Laboratories, Hercules, CA, USA). Target-gene expression was analyzed as the relative quantity of the target gene using the 2^−∆∆CT^ threshold cycle method compared to the glyceraldehyde-3-phosphate dehydrogenase as the reference gene used to normalize the gene expression data.

### 2.8. Statistical Analysis

All samples were evaluated in triplicate and the data are expressed as the mean ± standard deviation. One-way analysis of variance was performed using SPSS software (version 20.0; SPSS, Inc., Chicago, IL, USA). Post hoc tests were performed using Duncan’s test for multiple comparisons at a significance level of 0.05. Principal component analysis (PCA) was conducted and visualized using MetaboAnalyst version 5.0 software (20 May 2021, https://www.metaboanalyst.ca).

## 3. Results

### 3.1. Screening of CLA-Producing LAB 

An LA concentration of 0.5% at 37 °C inhibited the growth of the LA-tolerant LAB. However, when the LA concentration in the MRS agar medium decreased to 0.2%, the LAB formed colonies of various sizes surrounded by a halo on the medium (Appendix A). The isolates were classified according to halo size: 63 were large halo producers, 38 were medium producers, and 89 were low producers and subjected to additional screening by UV spectral scan analysis. Among the 63 selected isolates that formed large halos with large colonies, only 24 isolates exhibited characteristic absorption peaks at 228–235 nm, indicating the presence of conjugated double bonds (Appendix A). When the CLA productivity was measured at 233 nm, the 24 strains which exhibited a characteristic peak could produce above 50.0 μg/mL of CLA and the 10 LAB strains of JBCC105611, JBCC105117, JBCC105675, JBCC105634, JBCC105676, JBCC105674, JBCC105683, JBCC105655, JBCC105645, and JBCC105686 that produced more than 80.0 μg/mL CLA (from 86.0 to 125.5 μg/mL) were ultimately selected for further experiments (Table 1). 

### 3.2. Identification of CLA-Producing LAB 

According to the 16S rRNA sequencing, the analysis showed improved resolution in its results. The isolates JBCC105611, JBCC105634, and JBCC105655 showed high similarity to *Lactiplantibacillus paraplantarum* (99–100%). The isolates JBCC105675, JBCC105683, and JBCC105645 were identified as *Lactiplantibacillus plantarum*, and JBCC105676 and JBCC105674 were identified as *Lactiplantibacillus pentosus*. The JBCC105117 and JBCC105686 strains showed high identity with *Pediococcus acidilactici* (99%) and *Leuconostoc mesenteroides* (100%), respectively (Appendix A).

### 3.3. Isomeric Composition Analysis of CLA by GC 

To analyze the geometric isomeric ratio of *cis*-9, *trans*-11-CLA, and *trans*-10, *cis*-12-CLA, which show significantly different biological activities [37], GC analysis was performed, which revealed that all the selected strains showed different geometric isomeric ratios, as shown in Table 1. The *Lactobacillus plantarum* JBCC105683 and *L. pentosus* JBCC105676 strains produced the largest amounts of CLA (748.8 and 726.9 μg/mL, respectively) in the presence of 600 μg/mL of substrate, showing high conversion rates of 124.8% and 121.2%, respectively. They also showed high conversion rates, of approximately 40%, in the presence of 1000 μg/mL of LA. The lower conversion capacity to CLA at higher LA concentrations can be ascribed to the antimicrobial ability of LA [38]. In addition, different CLA isomer ratios were observed for each strain. In particular, *L. paraplantarum* JBCC105611, *P. acidilactici* JBCC105117, *L. paraplantarum* JBCC105634, *L. plantarum* JBCC105645, and *L. mesenteroides* JBCC105686 produced similar proportions of the two isomers (an approximately 50:50 ratio), whereas *L. plantarum* JBCC105675, *L. pentosus* JBCC105676, *L. pentosus* JBCC105674, *L. plantarum* JBCC105683, and *L. paraplantarum* JBCC105655 synthesized more *cis*-9, *trans*-11-CLA than *trans*-10, *cis*-12-CLA (approximately 80:20 ratio). 

### 3.4. pH and Bile Salt Tolerance of CLA-Converting LAB

As shown in Table 2, the 10 selected LAB strains showed relatively high survival rates at pH 2.0 for 2 h, even though the survival rate declined after 2 h. *L. paraplantarum* JBCC105634, *L. paraplantarum* JBCC105655, and *L. pentosus* JBCC105674 showed the highest acid tolerance. The bile tolerance of the selected CLA-producing LAB is shown in Table 3. Although the cell viability of the CLA-producing LAB slightly declined with the increasing bile concentration, nearly all the selected strains grew in 5% bile salts. 

### 3.5. Antimicrobial Activity

Previous studies have suggested that LAB in the gut microflora play an important role as a barrier against pathogenic strains in the GI tract and inhibit attachment to cultured uroepithelial cells, intestinal cells, and mucus [39]. As shown in Table 4, all the selected LAB strains inhibited the growth of all the selected pathogenic strains (Appendix A). In particular, strong antimicrobial activities against *S. aureus* were observed for all the selected LAB, showing nearly identical results to the control strain, *L. rhamnosus* GG. However, the capacity to inhibit pathogenic bacteria, such as *E. coli* and *B. cereus*, by all the selected LAB, was higher than that of the control strain, indicating their suitability as probiotics. Other pathogenic bacteria, including *S. epidermidis*, *P. aeruginosa*, and *P. putida* also showed similar inhibition activities compared to the control strains. The antimicrobial activity of the CLA-producing LAB isolated in this study inhibited both gram-positive and gram-negative pathogens more effectively than the control strain.

### 3.6. Intestinal Adhesion Ability of LAB Strains to Caco-2 Cells

To estimate the adhesion capacity of the high CLA-producing LAB, we examined their adhesion to the human epithelial cell line Caco-2. As shown in Figure 1, the 10 selected bacterial cells showed variable adhesion capacity depending on the strain. The isolates *L. plantarum* JBNU105683, *L. paraplantarum* JBCC105655, *L. plantarum* JBNU105645, and *L. paraplantarum* JBCC105634 showed slightly higher adhesion capacities to human epithelial cells (8.20–8.46 log CFU/mL) (*p* > 0.05) at levels comparable to those of the commercial probiotic strain, *L. rhamnosus* GG (8.17 log CFU/mL). *L. plantarum* JBCC105675, *L. mesenteroides* JBCC105686, *L. pentosus* JBCC105676, *L. pentosus* JBCC105674, and *L. paraplantarum* JBCC105611 exhibited intermediate (7.38–7.81 log CFU/mL) capacities. The *P. acidilactici* JBCC105117 strains exhibited low adhesion capacity (6.89 log CFU/mL). Thus, the adherence of the isolates in the GI tract may extend the residence time in the host.

### 3.7. Effects of LAB Strains on Growth and Cytokine Secretion in RAW 264.7 Cells

To determine the toxicity of the isolates, RAW 264.7 cells were treated with the LAB for 72 h and the cell viability was evaluated by the MTT assay. The treatment with *L. paraplantarum* JBCC105611, *L. paraplantarum* JBCC105634, and *L. mesenteroides* JBCC105686 resulted in similar cell viabilities compared to *L. rhamnosus* GG and other isolates, with no effects on cell toxicity (Appendix A). This indicates that none of the selected strains in this study were cytotoxic to RAW 264.7 cells. We demonstrated the production of different pro- and anti-inflammatory responses in murine macrophage RAW 264.7 to examine the immune-modulating functions of the heat-killed LAB. As shown in Figure 2, some strains showed similar cytokine production patterns, except for transforming growth factor-β. The three selected strains, *L. paraplantarum* JBCC105655, *L. plantarum* JBCC105683, and *L. pentosus* JBCC105676, induced a significantly higher secretion of pro-inflammatory cytokines TNF-α, IL-1β, IL-12, and IL-6 in RAW 264.7 cells compared to the positive control *L. rhamnosus* GG. IL-10, an anti-inflammatory cytokine, was also stimulated by isolated LAB compared to *L. rhamnosus* GG, except for the strains *L. paraplantarum* JBCC105634 and *L. plantarum* JBCC105645. In particular, *L. plantarum* JBCC105655, *L. plantarum* JBCC105683, and *P. acidilactici* JBCC105117 highly stimulated macrophage cells to simultaneously produce IL-10 and pro-inflammatory IL-12, exhibiting higher levels than those obtained with LPS.

### 3.8. Effects of LAB Strains on In Vivo Caenorhabditis Elegans Assay for Immune Response

Four strains, including *P. acidilactici* JBCC105117, *L. paraplantarum* JBCC105611, JBCC105655, and *L. plantarum* JBCC105683 strongly stimulated PMK-1 expression (Figure 3). When the transcript levels were measured in young adult *fer-15*;*fem-1* worms conditioned with four LAB strains for 24 h to quantitatively verify their immune activity, *P. acidilactici* JBCC105117, *L. paraplantarum* JBCC105655, and *L. plantarum* JBCC105683 strongly stimulated the expression of *clec-60*, which is consistent with the *pmk-1*::*GFP* assay (Figure 4). Unexpectedly, *L. paraplantarum* JBCC105611 showed very low expression of clec-60, despite the increased induction activity of pmk-1::GFP. The reason for this is still unclear, but one likely explanation is the presence of diverse factors related to the immune pathway differences, such as TNF-*α*, IL-1*β*, IL-12, and IL-6. In order to confirm our immune results from the nematode innate immune system, we examined the immune-modulating functions of heat-killed lactic acid bacteria in the murine macrophage cell line RAW 264.7.

### 3.9. PCA

To comprehensively evaluate the selected LAB strains, we assessed their relationships with the probiotic phenotypes (acid tolerance, bile tolerance, cell adhesion, antibiotic resistance, and antimicrobial activity), immune response factors (IL-1β and IL-10), and CLA productivity subjected to PCA. As illustrated in Figure 5, PC1 (variance 36.4%) was mainly related to CLA productivity, acid/bile tolerance, and cell adhesion ability. PC2 (variance 26.9%) expressed cytokine expression and antimicrobial activity. In particular, three strains, *L. paraplantarum* JBCC105655, *L. plantarum* JBCC105683, and *L. pentosus* JBCC105674, located in the right part of the PC1 plane, were characterized by high CLA productivity and high probiotic properties. The other groups, including *L. paraplantarum* JBCC105655, *L. plantarum* JBCC105683, and *P. acidilactici* JBCC105117, displayed high levels of IL-1β or IL-10 expression and antimicrobial activity. In particular, *L. paraplantarum* JBCC105655 and *L. plantarum* JBCC105683 significantly (*p* < 0.05) stimulated the expression of both pro- and anti-inflammatory cytokines.

## 4. Discussion

LAB are important producers of valuable CLAs that exert health-promoting effects. Recent studies have focused on producing functionally enhanced fermented foods and using CLA-producing strains as probiotic strains. In this study, we aimed to find CLA-converting LAB from *Jeot-gal*, a traditional Korean fermented seafood, due to its severe production environments with high salt content. We successfully isolated CLA-converting LAB by using systematic three-step screening approaches for LA tolerance, UV spectrum, and GC analysis and found 10 strains, including *L. paraplantarum* (3), *L. plantarum* (3), *L. pentosus* (2), *L. mesenteroids* (1), and *P. acidilactici* (1). Although LA serves as the substrate for CLA biosynthesis, LAB are generally intolerant to LA [40]. It has been suggested that the conversion of LA to CLA might be a detoxification mechanism in bacterial strains and positively correlated with CLA production [41,42]. Therefore, their adaptation at high LA concentrations could be used to identify high-level CLA-producing LAB with high LA tolerance for high-level CLA production.

In our study, most of the selected LAB, except for the *P. acidilactici* JBCC105117 and *L. mesenteroides* JBCC105686 strains, showed a tendency to decrease with the increasing concentrations of LA. The growth of LAB can be affected even at lower LA levels (25 μg/mL) [42]. However, our selected strains were able to resist much higher LA concentrations, although the growth of selected LAB was inhibited in a concentration-dependent manner. In particular, three selected strains, *L. paraplantarum* JBCC105611, *L. pentosus* JBCC105674, *and L. plantarum* JBCC105645 showed relatively high LA resistance and tolerated up to 200 μg/mL of LA (*p* < 0.05). The growth rate of *L. paraplantarum* JBCC105611 was markedly affected during the log phase of bacterial growth when the LA level increased to over 200 μg/mL. Previously, when the influence of LA on growth rate was examined at concentrations of 1, 100, 200, 600, and 1000 μg/mL for *L. plantarum* isolated from fermented Chinese pickles, viable cell counts of the strain were found to decline after 36 h [38]. Compared to these findings, viable cell counts of *L. paraplantarum* JBCC105611 were maintained until 48 h despite the presence of LA, indicating a relatively high tolerance to LA. CLA production by bacteria occurs in the stationary stage [38,43]. Interestingly, the growth pattern of *L. paraplantarum* JBCC105611 can maintain a stationary phase for a relatively long time, indicating that stable production of high-level CLA might be possible compared to other strains, resulting in the highest CLA productivity with a yield of approximately 125.5 μg/mL, which was followed by *P. acidilactici* JBCC105117 and *L. plantarum* JBCC105675 producing 106.1 and 95.8 μg/mL of CLA, respectively. To the best of our knowledge, this is the first report confirming CLA production by *L. paraplantarum* isolated from fermented foods. *L. plantarum* is one of the most widely studied LAB because of its high CLA-synthesis rate [38]. *L. plantarum* lp15 isolated from pickles could produce 72.2 μg/mL in the MRS medium containing 1 mg/mL of LA. However, *L. plantarum* JBCC105675 isolated from *Jeot-gal*s produced CLA at a yield of 95.8 μg/mL, even at a low LA concentration of 5 mg/mL, representing an improved production compared to *L. plantarum* lp15. These results clearly showed that *L. plantarum* JBCC105675 may exhibit a higher conversion rate than *L. plantarum* lp15. Therefore, the selected LAB with high conversion rates are valuable candidates for industrial applications. Unexpectedly, we found that CLA productivity at a low substrate concentration was above a 100%-conversion rate (Table 1). This can be explained with additional CLA production from unknown constituents supplied by the medium or from LA already present in the cell membrane. Indeed, it was reported that up to 20% of the LA can be found in the cell membrane of *L. acidophilus and L. helveticus* strains in stress conditions such as desiccation [44].

In this study, we found that the CLA isomer profiles varied for each strain. *L. paraplantarum* JBCC105611, *P. acidilactici* JBCC105117, *L. paraplantarum* JBCC105634, *L. plantarum* JBCC105645, and *L. mesenteroides* JBCC105686 generated similar proportions of the two isomers (approximately 50:50 ratio), whereas *L. plantarum* JBCC105675, *L. pentosus* JBCC105676, *L. pentosus* JBCC105674, *L. plantarum* JBCC105683, and *L. paraplantarum* JBCC105655 produced *cis*9, *trans*11-CLA as the main isomer (80% of total CLA), followed by *trans*10, *cis*12-CLA (20%), which is identical to *L. plantarum* lp15 from naturally fermented Chinese pickles (75:25) [38]. CLA isomer production by *L. plantarum* involves four enzymes that catalyze hydration/dehydration (CLA-HY), oxidation of the hydroxyl groups and reduction of the oxo groups (CLA-DH), migration of carbon-carbon double bonds (CLA-DC), and saturation of carbon-carbon double bonds [45,46]. A recent study showed that the *cis*9, *trans*11-CLA isomer accumulation by *L. plantarum* is significantly related to α-enolase, a multifunctional-anchorless-surface protein that plays a role in cell detoxification from polyunsaturated fatty acids, such as linoleic acid, along with the linoleate isomerase complex [47]. Nevertheless, the reason why bacteria exhibit different isomer metabolism remains unclear. *L. plantarum* NCUL005, isolated from natural sauerkraut, produced CLA isomers in a 30:70 ratio [29]. The *cis*9, *trans*11-CLA isomer showed a more potent antiproliferative effect on the viability of two cancer cell lines (SW480 and HT-29) than the c9, t11 isomer [48].

Resistance to low pH and bile salts in the acidic environment of the stomach and upper part of the intestine has been regarded as an important property in the assessment of probiotic strains. The low pH of 1.0 to 2.5 and bile concentrations of 0.3–0.4% are strong barriers against the entry of bacteria into the intestinal tract [49]. Tolerance under these conditions is a stronger discriminative parameter for selecting acid-tolerant bacteria for probiotics [50]. It seems that the high CLA-converting strains exhibited higher survival than other strains of *L. plantarum*, *L. acidophilus*, and *L. paracasei*, which showed an average survival rate of less than 40% after exposure to pH 2.0 [51,52]. In particular, *L. paraplantarum* JBCC105634, *L. paraplantarum* JBCC105655, and *L. pentosus* JBCC105674 exhibited survival rates above 50% after exposure to pH 2.0. Once the LAB reach the small intestinal tract, their bile resistance is crucial because the bacterial cell membrane, which consists of lipids and fatty acids, is very susceptible to the detergent-like characteristics of the bile salts [50]. Although the cell viability of the CLA-producing LAB slightly declined with an increase in bile concentration (*p* < 0.05), almost all the selected strains were able to tolerate the bile salts even at high concentrations. *P. acidilactici* JBCC105117 showed significantly high survival rates of 99.4% and 96.1% at 3% and 5% bile salts, respectively (*p* < 0.05).

In our study, we evaluated the selected LAB using a *Caenorhabditis elegans* surrogate in vivo model to determine their potential for immunity. Our results clearly showed that two strains, *L. paraplantarum* JBCC105655 and *L. plantarum* JBCC105683, induced a significantly higher secretion of TNF-α, IL-1β, IL-12, and IL-10 in RAW 264.7 cells, compared to *L. rhamnosus* GG as a positive control. In addition, *P. acidilactici* JBCC105117 stimulated macrophage cells to simultaneously produce IL-10 and pro-inflammatory IL-12. However, we could not observe any distinct expression of immune-modulating functions with *L. paraplantarum* JBCC105611, which clearly matched the results described above, except for the PMK-1 signaling test. Taken together, these three LAB strains showed distinct pro- and anti-inflammatory cytokine expression levels, indicating that they may play an important role in balancing the immune response. As we used not only heat-killed bacterial preparations in order to remove possible bacterial metabolite interruptions but also the nematode innate immune system of *C. elegans* in vivo, the observed immune functions of the three strains might be related to both the surface properties and the inner antimicrobial peptides. It is well known that the cell-wall structure of non-pathogenic gram-positive bacteria acts as an excellent inducer of immune responses, such as that against pathogenic gram-negative bacteria [53]. It was also shown that different LAB stimulate different levels of various cytokines, such as TNF-α, IFN-γ, and IL-12. Although we did not measure IFN-γ production by the three LAB strains directly in our studies, the selected strains must be effective for immune function as macrophage-derived IL-12 stimulates IFN-γ production in T cells and NK cells, which can activate the specific immune responses of the Th-2 pathway, as well as IgE secretion in mice [54]. In addition, the secretion of Th-2 cytokines plays a major role in the perpetuation of immunological responses in allergic diseases [55]. Interestingly, the cytokine IL-12 induction in the three strains was higher than in the controls, including LPS. Excessive production of IL-12 can impair the organ-specific auto-immunity balance. However, IL-10 suppresses IL-12 production and supports the function of the regulatory T cells. Their mutually antagonistic functions have been reported by several authors [56,57]. Thus, the balanced production of IL-10 and IL-12 by the three strains inhibits polarization of the immune response, which is related to the Th1-type immune response, whereas the Th2-type is important for host immunity. *L. plantarum* isolated from *Kimchi* has been reported to induce a macrophage-derived Th1 response, which may be helpful for anti-allergic effects in vitro [58]. The innate immune response serves not only as the first line of defense, but also plays a crucial role in the development of subsequent adaptive immune responses. Therefore, the release of pro- and anti-inflammatory cytokines from immune cells stimulated by selected lactic acid bacteria, *L. plantarum* JBCC105655, *L. plantarum* JBCC105683, and *P. acidilactici* JBCC105117, may modulate the innate and adaptive immune system and inflammatory response. In addition, the conjugated linoleic acid (CLA), which exhibits anti-inflammatory and anti-carcinogenic properties, produced by the bacterial strains in this study may have a positive effect on the intestinal environment. Furthermore, kinetic studies of each enzyme are needed in future investigations.

## 5. Conclusions

In this study, LAB from *Jeot-gal*s were identified and found to have diverse CLA production yields and isomer ratios. Moreover, these LAB showed potential probiotic properties, suggesting their potential as beneficial probiotics that produce CLA as a functional ingredient. *P. acidilactici* JBCC105117, *L. paraplantarum* JBCC105655, and *L. plantarum* JBCC105683 strongly stimulated the immunological regulatory gene PMK-1 and a host defense antimicrobial peptide gene, *clec-60,* in *C. elegans* and produced the significant induction of tumor necrosis factor-α, interleukin (IL)-1β, IL-6, IL-12, and IL-10 in RAW 264.7 macrophages, indicating that they are good candidates for probiotics with high CLA-converting activity. The LAB from *Jeot-gal*s are good candidates for manufacturing functionally enhanced fermented foods as functional starters with the beneficial effects of CLA.

## Figures and Tables

**Figure 1 microorganisms-09-02247-f001:**
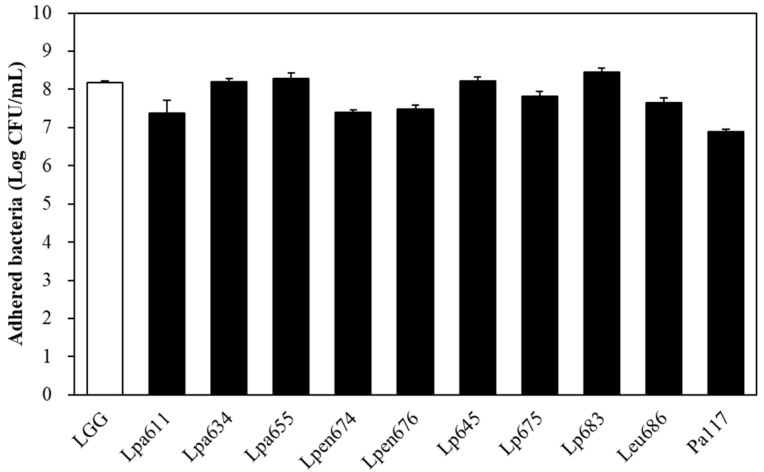
Adhesion capacity of *L. rhamnosus* GG and selected lactic acid bacteria strains to Caco-2 cells.

**Figure 2 microorganisms-09-02247-f002:**
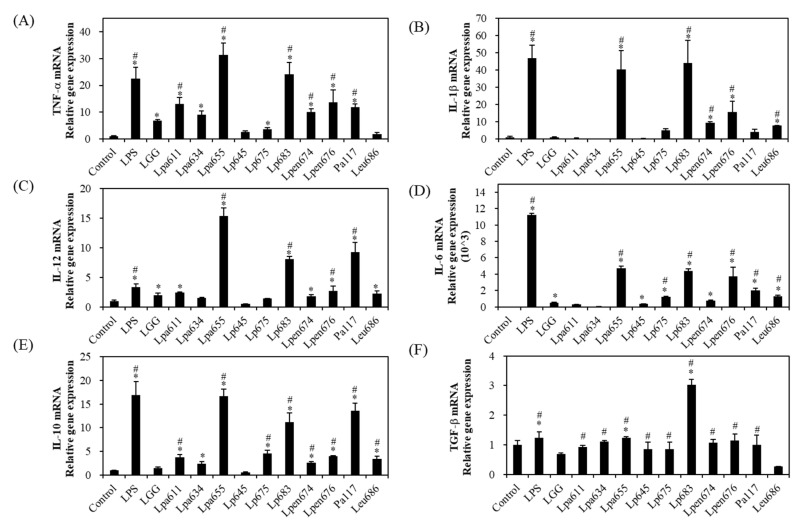
Pro-inflammatory (**A**–**D**) and anti-inflammatory cytokine (**E**,**F**) production by RAW 264.7 cells exposed to various heat-killed *L. rhamnosus* GG and selected lactic acid bacteria strains. RAW 264.7 cells were cultured in the medium (control), LPS (1 mg/mL), or heat-killed bacteria (8 log CFU/mL) for 48 h. *Lactocaseibacillus rhamnosus* GG, LGG; Pa, *Pediococcus acidilactici*; Lpa, *Lactobacillus paraplantarum*; Lp, *Lactobacillus plantarum*; Lpen, *Lactobacillus pentosus*; Leu, *Leuconostoc mesenteroids*. * *p* < 0.05 compared to the negative control; # *p* < 0.05 compared to LGG.

**Figure 3 microorganisms-09-02247-f003:**
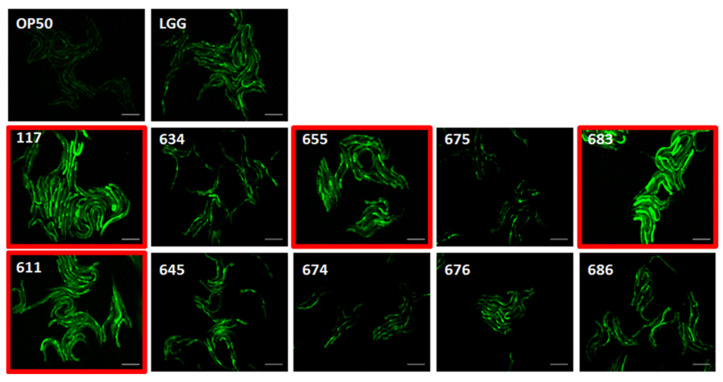
Induction of *pmk-1*::*GFP* exposed to selected LAB strains for 24 h. Images were acquired at the same time point with the same exposure time. *Escherichia coli* strain OP50 and *L. rhamnosus* GG are negative and positive controls, respectively. LGG, *Lactocaseibacillus rhamnosus* GG; 117, *Pediococcus acidilactici*; 611, 634, 655, *Lactobacillus paraplantarum*; 645, 675, 683, *Lactobacillus plantarum*; 674, 676, *Lactobacillus pentosus*; 686, *Leuconostoc mesenteroides*.

**Figure 4 microorganisms-09-02247-f004:**
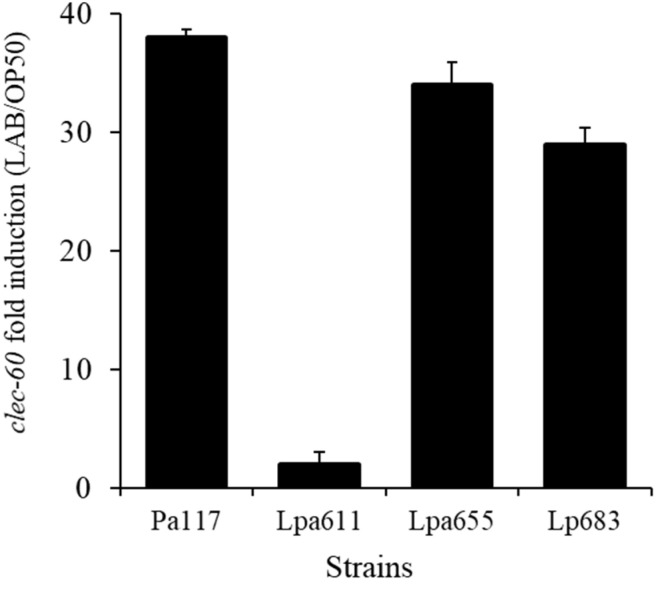
qRT-PCR analysis to evaluate the impact of conditioning with four LAB strains to *clec-60* associated with nematode immune responses. Transcript levels were measured in young adult *fer-15*; *fem-1* worms conditioned with four LAB strains for 24 h.

**Figure 5 microorganisms-09-02247-f005:**
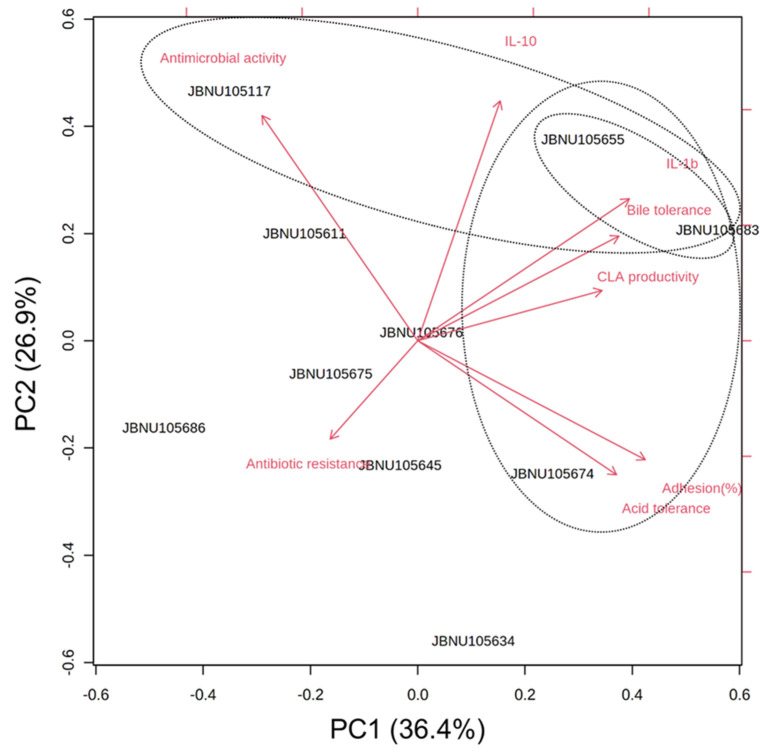
Principal component analysis (PCA) of six functional probiotics properties (acid tolerance, bile tolerance, antimicrobial activity, antibiotic resistance, and adhesion), CLA productivity, and cytokine expression% for 10 *Lactobacilli* strains isolated from *Jeot-gal*.

**Table 1 microorganisms-09-02247-t001:** Production of individual isomers and total CLA by LAB isolated from *Jeot-gal*s.

Strain	LA (μg/mL)	CLA Productivity					Conversion Rate (%)	Isomer Ratio
		*cis*9, *trans*11-CLA (μg/mL)	Conversion Rate (%)	*trans*10, *cis*12-CLA (μg/mL)	Conversion Rate (%)	Total CLA (μg/mL)		
*L. paraplantarum* JBNU105611	100	91.5 ± 1.9	91.5	91.1 ± 3.6	91.1	182.6 ± 5.5	182.6	50:50
	200	87.4 ± 3.8	43.7	87.3 ± 1.8	43.6	174.7 ± 5.6	87.3	50:50
	600	80.6 ± 5.8	13.4	77.1 ± 5.4	12.9	157.7 ± 11.1	26.3	51:49
	1000	75.2 ± 5.8	7.5	73.3 ± 0.0	7.3	148.5 ± 5.8	14.8	51:49
*L. paraplantarum* JBNU105634	100	73.8 ± 0.0	73.8	75.8 ± 0.0	75.9	149.7 ± 0.0	149.7	49:51
	200	77.9 ± 1.9	39.0	77.1 ± 1.8	38.6	155.0 ± 3.7	77.5	50:50
	600	76.5 ± 0.0	12.8	67.5 ± 1.8	11.3	146.0 ± 1.8	24.3	52:48
	1000	76.5 ± 3.8	7.7	74.6 ± 5.4	7.5	151.1 ± 9.2	15.1	51:49
*L. paraplantarum* JBNU105655	100	163.4 ± 3.8	163.4	86.0 ± 0.0	86.0	249.4 ± 3.8	249.4	66:34
	200	308.6 ± 1.9	154.3	89.8 ± 1.8	44.9	398.4 ± 0.1	199.2	77:23
	600	228.5 ± 3.8	38.1	80.9 ± 3.6	13.5	309.5 ± 7.4	51.6	74:26
	1000	140.3 ± 5.8	14.0	82.2 ± 5.4	8.2	222.5 ± 11.1	22.3	63:37
*L. pentosus* JBNU105674	100	231.3 ± 7.7	231.3	73.3 ± 0.0	73.3	304.6 ± 7.7	304.6	76:24
	200	354.8 ± 9.6	177.4	80.9 ± 3.6	40.5	435.7 ± 13.2	217.8	81:19
	600	516.3 ± 7.7	86.0	80.9 ± 0.0	13.5	597.2 ± 7.7	99.5	86:14
	1000	182.4 ± 11.5	18.2	51.7 ± 1.8	5.2	234.1 ± 13.3	23.4	78:22
*L. pentosus* JBNU105676	100	261.1 ± 11.5	261.1	75.8 ± 3.6	75.9	337.0 ± 15.1	337.0	77:23
	200	519.0 ± 11.5	259.5	92.4 ± 1.8	46.2	611.3 ± 13.3	305.7	85:15
	600	638.4 ± 3.8	106.4	88.6 ± 0.0	14.8	727.0 ± 3.8	121.2	88:12
	1000	357.5 ± 5.8	35.8	72.0 ± 1.8	7.2	429.5 ± 7.6	43.0	83:17
*L. plantarum* JBNU105645	100	117.3 ± 7.7	117.3	110.1 ± 5.4	110.2	227.4 ± 13.1	227.4	52:48
	200	96.9 ±61.9	48.5	88.6 ± 3.6	44.3	185.4 ± 5.5	92.7	52:48
	600	101.0 ± 0.0	16.8	91.1 ± 0.0	15.2	192.1 ± 0.0	32.0	53:47
	1000	90.1 ± 7.7	9.0	77.1 ± 5.4	7.7	167.2 ± 13.1	16.7	54:46
*L. plantarum* JBNU105675	100	266.5 ± 7.7	266.5	89.8 ± 1.8	89.8	356.4 ± 9.5	356.4	75:25
	200	361.5 ± 11.5	180.8	65.7 ± 3.6	32.8	427.2 ± 15.1	213.6	85:15
	600	185.1 ± 11.5	30.9	32.7 ± 0.0	5.4	217.8 ± 11.5	36.3	85:15
	1000	42.6 ± 1.9	4.3	27.6 ± 0.0	2.8	70.2 ± 2.0	7.0	61:39
*L. plantarum* JBNU105683	100	242.1 ± 0.0	242.1	92.4 ± 1.8	92.4	334.5 ± 1.8	334.5	72:28
	200	443.0 ± 3.8	221.5	110.2 ± 1.8	55.1	553.1 ± 5.6	276.6	80:20
	600	642.5 ± 1.9	107.1	106.3 ± 0.0	17.7	748.8 ± 2.0	124.8	86:14
	1000	316.8 ± 1.9	31.7	96.2 ± 0.0	9.6	412.9 ± 2.0	41.3	77:23
*L.mesenteroides* JBNU105686	100	101.0 ± 0.0	101.0	73.3 ± 0.0	73.3	174.3 ± 0.0	174.3	58:42
	200	90.1 ± 3.8	45.1	67.0 ± 1.8	33.5	157.1 ± 5.6	78.5	57:43
	600	83.3 ± 5.8	13.9	63.1 ± 3.6	10.5	146.5 ± 9.4	24.4	57:43
	1000	75.1 ± 9.6	7.5	60.6 ± 7.2	6.1	135.8 ± 16.8	13.6	55:45
*P. acidilactici* JBNU105117	100	73.8 ± 0.0	73.8	64.4 ± 1.8	64.4	138.2 ± 1.8	138.2	53:47
	200	80.6 ± 1.9	40.3	65.2 ± 3.6	32.6	148.8 ± 5.5	74.4	54:46
	600	73.8 ± 0.0	12.3	63.1 ± 0.0	10.5	137.0 ± 0.0	22.8	54:46
	1000	76.5 ± 0.0	7.7	64.4 ± 1.8	6.4	141.0 ± 1.8	14.1	54:46

Results are expressed with mean ± SD of duplicate experiments.

**Table 2 microorganisms-09-02247-t002:** Acid tolerance of CLA-producing LAB under acidic conditions (pH 2.0).

Strain	Viable Cell Counts (log CFU/mL)	
	Control	2 h
*L. paraplantarum* JBCC105611	8.91 ± 0.10	4.68 ± 0.24
*P. acidilactici* JBCC105117	9.12 ± 0.10	3.61 ± 0.16
*L. plantarum* JBCC105675	9.01 ± 0.04	4.02 ± 0.15
*L. paraplantarum* JBCC105634	7.88 ± 0.10	5.01 ± 0.53
*L. pentosus* JBCC105676	9.11 ± 0.21	4.31 ± 0.19
*L. pentosus* JBCC105674	9.05 ± 0.26	5.37 ± 0.22
*L. plantarum* JBCC105683	9.05 ± 0.04	5.03 ± 0.44
*L. paraplantarum* JBCC105655	8.97 ± 0.18	5.01 ± 0.51
*L. plantarum* JBCC105645	9.08 ± 0.04	4.99 ± 0.40
*L. mesenteroides* JBCC105686	8.77 ± 0.37	3.75 ± 0.19

**Table 3 microorganisms-09-02247-t003:** Viable cell counts (log CFU/mL) of selected CLA-producing LAB isolated from *Jeot-gal*s at different bile-salts concentrations.

Strain	Bile Concentration (%)				
	0	0.3	1	3	5
*L. paraplantarum* JBNU105611	8.91 ± 0.10 ^a^	8.89 ± 0.02 ^a^	8.02 ± 0.18 ^b^	7.16 ± 0.22 ^c^	7.08 ± 0.05 ^c^
*P. acidilactici* JBNU105117	9.12 ± 0.10 ^a^	9.03 ± 0.12 ^ab^	9.0 ± 0.11 ^ab^	8.19 ± 0.05 ^c^	7.89 ± 0.37 ^c^
*L. plantarum* JBNU105675	9.01 ± 0.04 ^a^	8.99 ± 0.06 ^b^	8.83 ± 0.04 ^c^	6.70 ± 0.13 ^d^	6.62 ± 0.07 ^d^
*L. paraplantarum* JBNU105634	7.88 ± 0.10 ^a^	7.81 ± 0.28 ^ab^	7.77 ± 0.13 ^b^	7.00 ± 0.18 ^c^	5.97 ± 0.21 ^d^
*L. pentosus* JBNU105676	9.11 ± 0.21 ^a^	8.93 ± 0.07 ^ab^	8.76 ± 0.01 ^c^	8.66 ± 0.11 ^c^	7.51 ± 0.22 ^d^
*L. pentosus* JBNU105674	9.05 ± 0.26 ^a^	8.89 ± 0.13 ^b^	8.17 ± 0.23 ^c^	5.74 ± 0.17 ^d^	5.35 ± 0.09 ^e^
*L. plantarum* JBNU105683	9.05 ± 0.04 ^a^	9.01 ± 0.08 ^a^	8.37 ± 0.21 ^b^	6.44 ± 0.29 ^c^	6.15 ± 0.29 ^c^
*L. paraplantarum* JBNU105655	8.97 ± 0.18 ^a^	8.07 ± 0.09 ^b^	6.68 ± 0.27 ^c^	6.13 ± 0.05 ^d^	5.17 ± 0.15 ^e^
*L. plantarum* JBNU105645	9.08 ± 0.04 ^a^	7.9 ± 0.23 ^b^	7.03 ± 0.08 ^c^	7.12 ± 0.08 ^c^	6.69 ± 0.17 ^c^
*L. mesenteroides* JBNU105686	8.77 ± 0.37 ^a^	8.55 ± 0.05 ^a^	7.61 ± 0.11 ^b^	7.02 ± 0.6 ^c^	6.218 ± 0.09 ^d^

Values in the same row with a.b.c.d.e are significant differences (*p* < 0.05) by ANOVA with Duncan’s multiple range test and results are expressed with mean ± SD of triplicate experiments.

**Table 4 microorganisms-09-02247-t004:** Antimicrobial test against pathogenic microorganisms of selected strains.

Strain	Pathogenic Microorganisms						
	*S. aureus*	*S. epidermidis*	*S. xylosus*	*P. aeruginosa*	*P. putida*	*B. cereus*	*E. coli*
*L. rhamnosus* GG *	++++	+	+++	++	++	++	+
*L. paraplantarum* JBNU105611	++++	++	+++	+++	+++	++++	++++
*P. acidilactici* JBNU105117	++++	+++	+++	+++	+++	++++	++++
*L. plantarum* JBNU105675	++++	+	+++	++	+++	++++	+++
*L. paraplantarum* JBNU105634	++++	++	+++	++	++	+++	+++
*L. pentosus* JBNU105676	++++	++	++++	++	+++	++++	++++
*L. pentosus* JBNU105674	++++	++	+++	++	+++	++++	++
*L. plantarum* JBNU105683	++++	+	+++	++	+++	++++	+++
*L. paraplantarum* JBNU105655	++++	+	+++	++	+++	++++	++++
*L. plantarum* JBNU105645	++++	+	+++	++	+++	++++	+++
*L. mesenteroides* JBNU105686	++++	++	+++	+++	++	+++	++++

* *L. rhamnosus* is used as control strain. ++++, inhibition zone > 4 mm; +++, inhibition zone > 3 mm; ++, inhibition zone > 2 mm; +, inhibition zone > 1 mm; <+, inhibition zone < 1 mm.

## Data Availability

The data presented in this study are available in the manuscript and Appendix A.

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
