# Peer review of "Probiotic Properties of Lactic Acid Bacteria with High Conjugated Linoleic Acid Converting Activity Isolated from Jeot-Gal, High-Salt Fermented Seafood"

_microorganisms, 2021, doi:10.3390/microorganisms9112247_

Round 1

Reviewer 1 Report

Thank you for your manuscript.

I have some concerns.

-I think that this study area is somehow overlapped previous studies. It seems to me that the approach is quite common. What is the novelty of this study?

-I am puzzled as to why there was need to screen antimicrobial activity, antibiotic resistance etc. if the topic of this manuscript is about the CLA converting strains. The title should be informative.

-Materials and Method and Result part must be improved. For example:

  • Line 90: Please add definition of MRS before the abbreviation.
  • Line 107: percentage of the isopropanol is unknown.
  • Line 136: If you are trying to show the gastric acid resistance, you might add pepsin (0.4%) to the acidic MRS broth.
  • Line 162: How many CFUs did you applied to the cell?
  • Line 222: "inhibited"
  • Line 224, 229, 346: If you have the data, then include it. If not, then re-write the sentence instead of using "data not shown".
  • Table 1: Please abbreviate "Linoleic acid". 
  • Table 4: It is not about the antibiotic resistance. Table 3 is duplicated!
  • Line 326: Here you wrote "heat inactive", but I could not find any "heat inactive" in the Materials and Methods section.
  • Line 393: You have already defined the abbreviation for linoleic acid, so remove it.
  • Line 426: Lp5 or Lp15?
  • Line 463-464: re-write the sentence.

-About the antibiotic resistance, I would suggest to use also E-test assay and compare the result with disk diffusion. (Because there is not Table 4, I cannot discuss it here anymore)

-I have question regarding cytokine secretion. Have you ever measured the cytokine with EILSA? Was there any difference between secretion and production?

-How much lactic acid (mM) was produced by these strains?  I think that actual concentration of the lactic acid produced by each of these strains could affect the antimicrobial activity.

Author Response

We thank editor and all reviewers for their thorough and constructive comments. We have addressed each comment in detail below, and the changes made to manuscript are highlighted in the text. We kindly request your reconsideration of our manuscript for publication in Microorganisms.

Referee: A

Thank you for your manuscript.

I have some concerns.

I think that this study area is somehow overlapped previous studies. It seems to me that the approach is quite common. What is the novelty of this study?

Response: We thank the reviewer for providing valuable comments and suggestions that have helped us improve the quality of our submitted manuscript. As we described in the title and manuscript, we tried to isolate novel CLA-producing LAB from Jeot-gals, which is a useful source for isolating probiotic strains due to the halotolerant and high cell viability under the low pH value. Moreover, there is no information about LAB having ability of CLA production from Jeot-gals and its probiotic properties. Therefore, in this study, we isolated LABs with producing CLA from Jeot-gals and assess the properties with respect to its potential use as probiotics. Thus, we evaluated the basic probiotic activities of isolates and immune modulation in vivo by C. elegans as well as in vitro by inflammatory cytokine assay.

I am puzzled as to why there was need to screen antimicrobial activity, antibiotic resistance etc. if the topic of this manuscript is about the CLA converting strains. The title should be informative.

Response: Linoleic acid, essential fatty acid of omega 6 series, should be provided as food or supplements because human cannot synthesize it naturally. Our purpose in this study is to find probiotic strain which specially having high CLA converting activity. Thus, when linoleic acid ingested in human intestine by dietary supplementation, CLA converting probiotics strains could convert linoleic acid to CLA which might result beneficial effect to human health. That is why we investigated both CLA converting acidity and as well probiotic properties. such as acid and bile tolerance, antibiotic susceptibility, antimicrobial activity and so on, are necessary for probiotic in human intestine.

Materials and Method and Result part must be improved. For example:

Line 90: Please add definition of MRS before the abbreviation.

Response: We added “de Man, Rogosa, and Sharpe (MRS)” before the abbreviation in line 95 and marked in the manuscript of the revised manuscript following the reviewer’s comments.

Line 107: percentage of the isopropanol is unknown.

Response: We thank the reviewer for this comments. Purity of isopropanol of “99.9%” was added in line 112 in response to the reviewer’s comment and high purity isopropanol used for fatty acid extraction with hexane (95% purity) (sample : isopropanol : hexane = 1 : 2 : 1.5) as described in line 113-115.

Line 136: If you are trying to show the gastric acid resistance, you might add pepsin (0.4%) to the acidic MRS broth.

Response: In probiotic property tests, pepsin is used for gastric acid resistance. However, in many studies, pH adjustment also used for acid tolerance to evaluate viability in the gastrointestinal tract as following references. According to Xu et al. (2020), the pH of gastric juice is typically 3.0, and pH 2.0 is often used to simulate extreme stomach conditions.

Pradhan, P., & Tamang, J. P. (2021). Probiotic properties of lactic acid bacteria isolated from traditionally prepared dry starters of the Eastern Himalayas. World Journal of Microbiology and Biotechnology, 37(1), 1-13.

Won, S. M., Chen, S., Park, K. W., & Yoon, J. H. (2020). Isolation of lactic acid bacteria from kimchi and screening of Lactobacillus sakei ADM14 with anti-adipogenic effect and potential probiotic properties. LWT, 126, 109296.

Xu, Y., Zhou, T., Tang, H., Li, X., Chen, Y., Zhang, L., & Zhang, J. (2020). Probiotic potential and amylolytic properties of lactic acid bacteria isolated from Chinese fermented cereal foods. Food Control, 111, 107057.

Line 162: How many CFUs did you applied to the cell?

Response: We thank the reviewer for this comments. For the adherence assay, the Caco-2 cell monolayers were incubated with 1 × 108–1 × 109 CFU/mL of bacteria at 37 °C for 2 h. We added and marked in the manuscript of the revised manuscript in line 168.

Line 222: "inhibited"

Response: Thanks for pointing out the spelling error. We changed “inhibited” in line 228 and marked in the manuscript of the revised manuscript.

Line 224, 229, 346: If you have the data, then include it. If not, then re-write the sentence instead of using "data not shown".

Response: We added Supplementary Figure S1, Figure S2, and Figure S5 and marked in the manuscript of the revised manuscript in line 230, 235 and 360, respectively, in response to the reviewer’s comment.

Table 1: Please abbreviate "Linoleic acid". 

Response: We changed “Linoleic acid” to “LA” in Table 1 and marked in the manuscript of the revised manuscript.

Table 4: It is not about the antibiotic resistance. Table 3 is duplicated!

Response: We thank the reviewer for this critical comments and apologize for the imprecise data in the manuscript. We added correct Table 4 in the manuscript in line 315-318.

Line 326: Here you wrote "heat inactive", but I could not find any "heat inactive" in the Materials and Methods section.

Response: We thank the reviewer for this comments. We added “heat-killed LAB (100 °C, 30 min)” in line 207.

Line 393: You have already defined the abbreviation for linoleic acid, so remove it.

Response: We removed “linoleic acid” in line 408.

Line 426: Lp5 or Lp15?

Response: We changed “lp5” to “lp15” in line 443.

Line 463-464: re-write the sentence.

Response: We changed the sentence like this:

 “The immune system of the intestinal barrier are consists of a combination of mucus, intestinal epithelial cells, immunoglobulin A (IgA), antimicrobial peptides, and other immune cells.

About the antibiotic resistance, I would suggest to use also E-test assay and compare the result with disk diffusion. (Because there is not Table 4, I cannot discuss it here anymore)

Response: Assessment of resistance and susceptibility to antibiotics was studied by a disk diffusion method according to modified method of Gupta et al. (2021). The Kirby-Bauer test, known as the disk-diffusion method, is the most widely used antibiotic susceptibility of isolated LAB using antibiotic discs. Results were interpreted and the resistant strains were selected after being compared with known standard given by the Clinical and Laboratory Standards Institute (CLSI) for antimicrobial susceptibility testing (Clinical and Laboratory Standards Institute 2012). The isolates with a zone of inhibition less than 14 mm were considered as resistant. These sensitive and resistance patterns were identified based on CLSI guidelines.

E-test is one of the antibiotic resistance assay, but antibiotic susceptibility assay using antibiotic discs are also widely used as following references.

Gupta, S., Mohanty, U., & Majumdar, R. K. (2021). Isolation and characterization of lactic acid bacteria from traditional fermented fish product Shidal of India with reference to their probiotic potential. LWT, 146, 111641.

Kim, H., Shin, M., Ryu, S., Yun, B., Oh, S., Park, D. J., & Kim, Y. (2021). Evaluation of Probiotic Characteristics of Newly Isolated Lactic Acid Bacteria from Dry-Aged Hanwoo Beef. Food Science of Animal Resources, 41(3), 468.

Wang, X., Wang, W., Lv, H., Zhang, H., Liu, Y., Zhang, M., ... & Tan, Z. (2021). Probiotic potential and wide-spectrum antimicrobial activity of lactic acid bacteria isolated from infant feces. Probiotics and antimicrobial proteins, 13(1), 90-101.

Sohn, H., Chang, Y. H., Yune, J. H., Jeong, C. H., Shin, D. M., Kwon, H. C., ... & Han, S. G. (2020). Probiotic Properties of Lactiplantibacillus plantarum LB5 Isolated from Kimchi Based on Nitrate Reducing Capability. Foods, 9(12), 1777.

Sharma, P., Tomar, S. K., Sangwan, V., Goswami, P., & Singh, R. (2016). Antibiotic resistance of Lactobacillus sp. isolated from commercial probiotic preparations. Journal of Food Safety, 36(1), 38-51.

Clinical and Laboratory Standards Institute. 2012. Performance Standards for Antimicrobial Susceptibility Testing: Twenty- Second Informational Supplement. CLSI Document M100-S22, Clinical Laboratory Standard Institute, Wayne, PA.

I have question regarding cytokine secretion. Have you ever measured the cytokine with EILSA? Was there any difference between secretion and production?

Response: In this study, we measured cytokines level in two ways. Cytokines in protein level using ELISA and gene expression in mRNA level using Quantitative reverse-transcription PCR (qRT-PCR). Two cytokines, TNF-a and IL-6, showed similar patterns of production and expression at protein and mRNA level, respectively, in RAW 264.7 cell. After that, we compared relative quantity of six pro- and anti-inflammatory cytokines target gene expression compared to GAPDH reference gene using qRT-PCR at the same time as shown in Figure 2.

How much lactic acid (mM) was produced by these strains?  I think that actual concentration of the lactic acid produced by each of these strains could affect the antimicrobial activity.

Response: We thank the reviewer for providing valuable comments and suggestions. The disc diffusion method was used in this study to determine the wide-spectrum antimicrobial activity of the LAB strains and their inhibitory capacity against intestinal opportunistic pathogens. Therefore, actual lactic acid concentration (mM) was not measured in this study. Lactic acid produced by lactic acid bacteria may reduce the pH and could be affected antimicrobial activity. However, we did not consider the lactic acid or influence of pH on the antimicrobial activity in this study.

Reviewer 2 Report

Song et al., have identified LABs that have prominent probiotic activity in terms of conversion of LA to CLA. Overall, the manuscript is well-written, and evidence supporting the conclusion has been presented. I have few comments as listed below:

  1. The introduction should contain a brief introduction about Jeot-gals, as many people might not know about it. A brief about what is the starting component and how it is prepared would let readers know what is Jeot-gal, although the authors have mentioned this in the discussion section.
  2. If the authors can, please add representative data of a. halo formation on MRS supplemented with LA b. absorption peaks at 228-235 nm to the supplementary information.
  3. The meaning of a b c and their composition is not clear. The provided table legend should be clear. For instance, authors can compare the values with that of WT in each row and mark with one asterisk or two asterisks and so on depending upon the p values obtained.
  4. Table 2: It would be better and easy to understand at a glance if the authors can present this data as a figure.
  5. Tables 3 and 4 have been duplicated and the data for antibiotic susceptibility is missing. Please provide the correct table 4.
  6. Line 316-319: This statement can be discussed if the authors have tested the ability of L. rhamnosus regarding the conversion efficiency of LA to CLA of this strain or if previously reported literature can be cited. Otherwise, it will be an overstatement and should be removed.
  7. Table 5: If the authors can show representative zones of inhibition pictures of the plates in the supplementary information, that would be better.
  8. Figure 1: Please provide more information regarding statistical analysis used in the figure, for example, please mention statistically significant compared to which sample was obtained, if multiple comparison was made, please specify.
  9. I think it is better to move figures 4 and 5 to the results section rather than in the discussion part.

Author Response

We thank editor and all reviewers for their thorough and constructive comments. We have addressed each comment in detail below, and the changes made to manuscript are highlighted in the text. We kindly request your reconsideration of our manuscript for publication in Microorganisms.

Referee: B

Song et al., have identified LABs that have prominent probiotic activity in terms of conversion of LA to CLA. Overall, the manuscript is well-written, and evidence supporting the conclusion has been presented. I have few comments as listed below:

Response: We thank the reviewer for providing valuable comments and suggestions that have helped us improve the quality of our submitted manuscript. We have addressed each comment in detail below, and the changes made to manuscript are highlighted in the text.

  1. The introduction should contain a brief introduction about Jeot-gals, as many people might not know about it. A brief about what is the starting component and how it is prepared would let readers know what is Jeot-gal, although the authors have mentioned this in the discussion section.

Response: According to the reviewer’s comment, we added brief information of Jeot-gal in Introduction part in line 79-83.

  1. If the authors can, please add representative data of a. halo formation on MRS supplemented with LA b. absorption peaks at 228-235 nm to the supplementary information.

Response: We added Supplementary Figure S1 and Figure S2 and marked in the manuscript of the revised manuscript in line 230 and 235, respectively, in response to the reviewer’s comment.

  1. The meaning of a b c and their composition is not clear. The provided table legend should be clear. For instance, authors can compare the values with that of WT in each row and mark with one asterisk or two asterisks and so on depending upon the p values obtained.

Response: In Table 3, the superscript letters were changed to one asterisk in the significantly high levels depending upon the p values obtained significant differences (p<0.05) by ANOVA with Duncan’s multiple range test in response to the reviewer’s comment.

  1. Table 2: It would be better and easy to understand at a glance if the authors can present this data as a figure.

Response: We tried to express the acid tolerance data in respond to the experiment protocol. Therefore, both initial viable count (log CFU/mL) and survived viable count after 2 hours at pH 2 which the result value of the survival rate is calculated was described in the Table.

  1. Tables 3 and 4 have been duplicated and the data for antibiotic susceptibility is missing. Please provide the correct table 4.

Response: We thank the reviewer for these critical comments and apologize for the imprecise data in the manuscript. We added correct Table 4 in the manuscript in line 315-318.

  1. Line 316-319: This statement can be discussed if the authors have tested the ability ofrhamnosus regarding the conversion efficiency of LA to CLA of this strain or if previously reported literature can be cited. Otherwise, it will be an overstatement and should be removed.

Response: We thank the reviewer for bringing this to our attention. According to the reviewer’s comment, we deleted an overstatement part and revised the sentence “The antimicrobial activity of CLA-producing LAB isolated in this study may strongly affect probiotic properties because the strains inhibited both gram-positive and gram-negative pathogens more effectively than the control strain.” in the revised manuscript in line 329-332.

Revised sentence: “The antimicrobial activity of CLA-producing LAB isolated in this study inhibited both gram-positive and gram-negative pathogens more effectively than the control strain.”

  1. Table 5: If the authors can show representative zones of inhibition pictures of the plates in the supplementary information, that would be better.

Response: We added Supplementary Figure S4 and marked in the manuscript of the revised manuscript in line 323 in response to the reviewer’s comment.

  1. Figure 1: Please provide more information regarding statistical analysis used in the figure, for example, please mention statistically significant compared to which sample was obtained, if multiple comparison was made, please specify.

Response: As we mentioned in the Figure 1 legend, multiple comparison was not made and asterisk above the bars indicate significant differences among groups (11 strains) (p<0.05). Therefore, LGG, Lpa634, and Lp683 strains significantly (p<0.05) higher than other eight strains. Also, we described this in the manuscript as follow: “The isolates L. paraplantarum JBNU105634 and L. plantarum JBNU105683 showed strong adhesion activity to human epithelial cells (11.2–11.5%) (p<0.05) at levels comparable to those of the commercial probiotic strain, L. rhamnosus GG (12.2%).” in line 341-343.

  1. I think it is better to move figures 4 and 5 to the results section rather than in the discussion part.

Response: Thanks for your consideration. But Since Figures must be embedded in the main text close to their result, thus, We did not changed the position of Figure 4 and 5 and left them in Discussion part in which their results are described.